# A Note on the Relativistic Transformation Properties of Quantum Stochastic Calculus

**DOI:** 10.3390/e27050529

**Published:** 2025-05-15

**Authors:** John E. Gough

**Affiliations:** Department of Physics, Aberystwyth University, Aberystwyth SY23 3QR, UK; jug@aber.ac.uk

**Keywords:** quantum Itō calculus, relativistic fields, Davies–Unruh effect

## Abstract

We present a simple argument to derive the transformation of the quantum stochastic calculus formalism between inertial observers and derive the quantum open system dynamics for a system moving in a vacuum (or, more generally, a coherent) quantum field under the usual Markov approximation. We argue, however, that, for uniformly accelerated open systems, the formalism must break down as we move from a Fock representation over the algebra of field observables over all of Minkowski space to the restriction regarding the algebra of observables over a Rindler wedge. This leads to quantum noise having a unitarily inequivalent non-Fock representation: in particular, the latter is a thermal representation at the Unruh temperature. The unitary inequivalence is ultimately a consequence of the underlying flat noise spectrum approximation for the fundamental quantum stochastic processes. We derive the quantum stochastic limit for a uniformly accelerated (two-level) detector and establish an open system description of the relaxation to thermal equilibrium at the Unruh temperature.

## 1. Introduction

The quantum stochastic calculus was introduced by Hudson and Parthasarathy [1] and Gardiner and Collett [2] as a technique for describing open quantum systems. Typically, the quantum system is considered to be at rest, with quantum noise entering as quantum input processes (quantum white noises). The question of a relativistically covariant formulation was addressed and solved by Frigerio and Ruzzier [3]. Their construction makes extensive use of unitary representations of the Poincaré group on the one-particle space for the input noise. However, we present a shorter and more direct argument based on quantum white noises.

In Section 2, we provide a direct argument that re-derives the transformation rules for the quantum Itō table for quantum stochastic calculus when we progress from one inertial frame to another.

One envisages a quantum mechanical system moving along a world line. Here, we adopt a semi-classical view that the position of the system is resolved only on length scales much larger than its de Broglie wavelength: in particular, its quantum mechanical nature is described by internal degrees of freedom only (specifically excluding its position and momentum observables), for which the underlying Hilbert space is $h_{0}$. In Section 2.4, we consider quantum open system models where we have a unitary evolution for the system and background field. We also consider coherent state fields.

In Section 3, we consider the Unruh effect for the case of uniformly accelerated systems. Here, we disagree with the claim of [3] that the transformation rules apply without modification to arbitrary moving observers. In fact, the non-Fock nature of the noise for uniformly accelerated observers means that the gauge process is not well defined. We shall, however, derive and solve the master equation in this regime. We set up the problem as a weak coupling limit in Section 5 and derive the open system description in terms of quantum stochastic evolutions. We outline the convergence in Section 6.

## 2. Quantum Stochastic Calculus for Inertial Observers

Let *K* and $K^{\prime }$ be inertial frames in a standard configuration with spacetime coordinates related by the Lorentz transformation(1)$$\begin{array}{c} t^{\prime }=\gamma \left(u\right)\left(t-\frac{u}{c^{2}}x\right),\;x^{\prime }=\gamma \left(u\right)\left(x-ut\right),\;y^{\prime }=y,\;z^{\prime }=z, \end{array}$$
where $\gamma \left(u\right)=1/\sqrt{1-\frac{u^{2}}{c^{2}}}$ (the interpretation is that $K^{\prime }$ moves along the *x*-axis of *K* with velocity *u*). In the following, we will ignore the other two space dimensions.

A system is assumed to follow a world line given by x=vt in *K*: that is, it moves with velocity *v* along the *x*-axis in *K* and therefore is seen by $K^{\prime }$ to move along the $x^{\prime }$-axis with velocity(2)$$\begin{array}{c} v^{\prime }=\frac{v-u}{1-\frac{uv}{c^{2}}}. \end{array}$$If we consider two events along the world line of the system, then the proper time Δτ that elapses in the rest frame of the system is related to the times Δt and $\Delta t^{\prime }$ measured in *K* and $K^{\prime }$, respectively, by(3)$$\begin{array}{c} \Delta \tau =\sqrt{1-\frac{v^{2}}{c^{2}}}\Delta t=\sqrt{1-\frac{v^{\prime 2}}{c^{2}}}\Delta t^{\prime }. \end{array}$$In other words, $\Delta \tau =\frac{1}{\gamma \left(v\right)}\Delta t=\frac{1}{\gamma \left(v^{\prime }\right)}\Delta t^{\prime }$. It follows that(4)$$\begin{array}{c} \Delta t^{\prime }=\zeta \left(u,v\right)\Delta t \end{array}$$
where $\zeta \left(u,v\right)=\gamma \left(v^{\prime }\right)/\gamma \left(v\right)$, and, after a little algebra, one obtains(5)$$\begin{array}{c} \zeta \left(u,v\right)=\gamma \left(u\right)\left(1-\frac{uv}{c^{2}}\right). \end{array}$$

Note that, if u≡v, then $K^{\prime }$ is co-moving with the system, so $\Delta t^{\prime }\equiv \Delta \tau$. Here, $\zeta \left(v,v\right)\equiv 1/\gamma \left(v\right)$, which is the correct time dilation factor between Δτ and Δt.

### 2.1. Fundamental Quantum Stochastic Processes

We begin with the standard Fock space construction. Let $\Gamma \left(h\right)={⨁}_{n=0}^{\infty }{⊗}_{\mathrm{symm}.}^{n}h$ be the (Bose) Fock over a separable Hilbert space h. The exponential vectors are defined as $|\exp\left(f\right)\rangle =1⊕f⊕\frac{f⊗f}{\sqrt{2!}}⊕\cdots$, and they form a total set in Γ(h). The Fock vacuum is then Φ=|exp(0)〉=1⊕0⊕0⊕⋯. As is well known, we have the functorial property $\Gamma \left(h_{1}⊕h_{2}\right)≅\Gamma \left(h_{1}\right)⊗\Gamma \left(h_{2}\right)$; see, for instance, [4].

The annihilator with test function g∈h is the operator defined as B(g)|exp(f)〉=〈g|f〉|exp(f)〉, and $B{\left(f\right)}^{*}$ is its adjoint. For *U* unitary, its second quantization Γ(U) by Γ(U)|exp(f)〉=|exp(Uf)〉, and we define the differential second quantization dΓ(H) of a self-adjoint operator *H* as the Stone generator of $\Gamma (e^{itH})$.

For the Hudson–Parthasarathy quantum stochastic calculus [1,4], one considers the Fock space $h=L^{2}\left(R_{+},dt\right)$, and the annihilation B(t) and creation $B^{*}\left(t\right)$ processes are given by the annihilation and creation processes with test function $1_{\left[0,t\right]}$ (the characteristic function of the time interval [0,t]). Likewise, the gauge process Λ(t) is the differential second quantization of the operator on $L^{2}\left(R_{+},dt\right)$, corresponding to pointwise multiplication by [0,t].

The functorial property implies the continuous tensor product decomposition $\Gamma \left(L^{2}\left(R_{+},dt\right)\right)=\Gamma (L^{2}\left(\left[0,t\right),dt\right)⊗\Gamma \left(L^{2}\left(\left(t,\infty \right],dt\right)\right)$ for each t>0. The factors may be viewed as the past and future Fock spaces at time *t*. A process X(t) is adapted if it acts trivially on the future factor $\Gamma (L^{2}\left(\left(t,\infty \right],dt\right))$ for each *t*. Hudson and Parthasarathy constructed a quantum Itō calculus based on future pointing increments (that is, nontrivial only on the future factor at each time).

### 2.2. Quantum White Noise for a Fixed Inertial Observer

We may alternatively construct the quantum stochastic calculus from quantum white noises. In the rest frame of the system, we introduce annihilation and creation operators $\bar{b}\left(\tau \right)$ and ${\bar{b}}^{*}\left(\tau \right)$ satisfying the singular commutation relations(6)$$\begin{array}{c} \left[\bar{b}\left(\tau \right),{\bar{b}}^{*}\left(\tau^{\prime }\right)\right]=\delta \left(\tau -\tau^{\prime }\right). \end{array}$$The quantum stochastic processes may then be defined as(7)$$\begin{array}{ccc} \left(\mathrm{The}\mathrm{Gauge}\mathrm{Process}\right)\;\bar{\Lambda }\left(\tau \right) & = & \int_{0}^{\tau }{\bar{b}}^{*}\left(\sigma \right)\bar{b}\left(\sigma \right)d\sigma , \end{array}$$(8)$$\begin{array}{ccc} \left(\mathrm{The}\mathrm{Annihilation}\mathrm{Process}\right)\;\bar{B}\left(\tau \right) & = & \int_{0}^{\tau }\bar{b}\left(\sigma \right)d\sigma , \end{array}$$(9)$$\begin{array}{ccc} \left(\mathrm{The}\mathrm{Creation}\mathrm{Process}\right)\;{\bar{B}}^{*}\left(\tau \right) & = & \int_{0}^{\tau }{\bar{b}}^{*}\left(\sigma \right)d\sigma . \end{array}$$This leads to the quantum Itō table [1](10)$$\begin{array}{c} \begin{array}{cc} d\bar{\Lambda }\left(\tau \right)d\bar{\Lambda }\left(\tau \right)=d\bar{\Lambda }\left(\tau \right), & d\bar{B}\left(\tau \right)d\bar{\Lambda }\left(\tau \right)=d\bar{B}\left(\tau \right)\\ d\bar{\Lambda }\left(\tau \right)d{\bar{B}}^{*}\left(\tau \right)=d\bar{B}\left(\tau \right), & d\bar{B}\left(\tau \right)d{\bar{B}}^{*}\left(\tau \right)=d\tau . \end{array} \end{array}$$

### 2.3. Relativistic Transformations Between Inertial Frames

In order to work out the corresponding processes in the inertial frame *K*, we observe that the process $\bar{\Lambda }\left(\tau \right)$ should count noise quanta along the world line from proper time 0 up to proper time τ. This must, of course, be observer-independent, so we identify $\bar{\Lambda }\left(\tau \right)$ with the process $\Lambda \left(t\right)=\int_{0}^{t}b^{*}\left(s\right)b\left(s\right)ds$. Therefore, ${\bar{b}}^{*}\left(\tau \right)\bar{b}\left(\tau \right)d\tau \equiv b^{*}\left(t\right)b\left(t\right)dt$, and, using $dt=\gamma \left(v\right)d\tau$, we deduce that(11)$$\begin{array}{c} b\left(t\right)\equiv \frac{1}{\sqrt{\gamma \left(v\right)}}\;\bar{b}\left(\tau \right). \end{array}$$

Therefore, $dB\left(t\right)=b\left(t\right)dt=\sqrt{\gamma \left(v\right)}\;\bar{b}\left(\tau \right)d\tau \equiv \sqrt{\gamma \left(v\right)}d\bar{B}\left(\tau \right)$. The full set is(12)$$\begin{array}{c} \begin{array}{cc} d\Lambda \left(t\right)=d\bar{\Lambda }\left(\tau \right), & dB\left(t\right)=\sqrt{\gamma \left(v\right)}\;d\bar{B}\left(\tau \right),\\ dB^{*}\left(t\right)=\sqrt{\gamma \left(v\right)}\;d{\bar{B}}^{*}\left(\tau \right), & dt=\gamma (v)\;d\tau . \end{array} \end{array}$$The corresponding Itō table in inertial frame *K* is readily deduced to be(13)$$\begin{array}{c} \begin{array}{cc} d\Lambda \left(t\right)d\Lambda \left(t\right)=d\Lambda \left(t\right), & dB\left(t\right)d\Lambda \left(t\right)=dB\left(t\right),\\ d\Lambda \left(t\right)dB^{*}\left(t\right)=dB^{*}\left(t\right), & dB\left(t\right)dB^{*}\left(t\right)=dt. \end{array} \end{array}$$

For instance, $dB\left(t\right)d\Lambda \left(t\right)=\sqrt{\gamma \left(v\right)}d\bar{B}\left(\tau \right)d\bar{\Lambda }\left(\tau \right)=\sqrt{\gamma \left(v\right)}d\bar{B}\left(\tau \right)=dB\left(t\right)$, etc.

The transformation law from *K* to $K^{\prime }$ is then easily obtained by noting that(14)$$\begin{array}{c} b^{\prime }\left(t^{\prime }\right)=\sqrt{\gamma \left(v\right)/\gamma \left(v^{\prime }\right)}\;b\left(t\right)=\frac{1}{\sqrt{\zeta \left(u,v\right)}}\;b\left(t\right). \end{array}$$This yields the transformation(15)$$\begin{array}{c} \begin{array}{cc} d\Lambda^{\prime }=d\Lambda , & dB^{\prime }=\sqrt{\zeta \left(u,v\right)}\;dB,\\ dB^{\prime *}=\sqrt{\zeta \left(u,v\right)}\;dB^{*}, & dt^{\prime }=\zeta \left(u,v\right)dt. \end{array} \end{array}$$In [3], the calculations are presented in terms of hyperbolic angles α,β, where $\tanh\alpha =-\frac{u}{c}$ and $\tanh\beta =\frac{v}{c}$. They obtain the transformations (Equation 15) with ζ=coshα+βsinhα; however, this by inspection agrees with our (Equation 5).

**Remark** **1**(**Fermi Noise**). *The above arguments can be applied to derive the transformation rules for fermionic fields without additional work. The Fermi [5] version can be constructed from anti-commuting processes satisfying $\{a\left(t\right),a^{*}\left(s\right)\}=\delta \left(t-s\right)$. It follows that the Fermi processes will again transform in the same way, i.e., Equations (Equation 12) and (Equation 15), as their Bose counterparts.*

### 2.4. Unitary Evolutions

We recall that the standard form of a unitary quantum stochastic differential equation describing our system (with underlying Hilbert space $h_{0}$) interacting with the noise is [1](16)$$\begin{array}{c} d\bar{U}\left(\tau \right)=d\bar{G}\left(\tau \right)\;\bar{U}\left(\tau \right),\;\bar{U}\left(0\right)=I, \end{array}$$
where(17)$$\begin{array}{c} d\bar{G}\left(\tau \right)=\left(\bar{S}-I\right)⊗d\bar{\Lambda }\left(\tau \right)+\bar{L}⊗d{\bar{B}}^{*}\left(\tau \right)-{\bar{L}}^{*}\bar{S}⊗d\bar{B}\left(\tau \right)-\left(\frac{1}{2}{\bar{L}}^{*}\bar{L}+i\bar{H}\right)⊗d\tau . \end{array}$$The operators $\bar{S},\bar{L},\bar{H}$ on $h_{0}$ are assumed to be unitary, bounded, and bounded self-adjoint, respectively.

The picture for reference frame *K* will be described by the same equations as (Equation 16) and (Equation 17) with the bars removed from the noise operators, with the proper time τ replaced by *t*: dU(t)=dG(t)U(t),U(0)=I,$$\begin{array}{c} dG\left(t\right)=\left(S-I\right)⊗d\Lambda \left(t\right)+L⊗dB^{*}\left(t\right)-L^{*}S⊗dB\left(t\right)-\left(\frac{1}{2}L^{*}L+iH\right)⊗dt \end{array}$$
and the new coefficient operators are as follows(18)$$\begin{array}{c} S=\bar{S},\;L=\frac{1}{\sqrt{\gamma (v)}}\;\bar{L},\;H=\frac{1}{\gamma (v)}\;\bar{H}. \end{array}$$A quantum dynamical semi-group ${\left({\bar{\Phi }}_{\tau }\right)}_{\tau \geq 0}$ on operators of $h_{0}$ is defined by(19)$$\begin{array}{c} \langle \psi_{1}|{\bar{\Phi }}_{\tau }\left(X\right)|\psi_{2}\rangle \;=\;\langle \psi_{1}⊗\mathrm{vac}|{\bar{U}}^{*}\left(\tau \right)\left(X⊗I\right)\bar{U}\left(\tau \right)|\psi_{2}⊗\mathrm{vac}\rangle , \end{array}$$
where |vac〉 is the Fock vacuum state.

Its Lindblad generator [6] is(20)$$\begin{array}{c} \bar{L}\left(X\right)=-i\left[X,\bar{H}\right]+\frac{1}{2}\left[{\bar{L}}^{*},X\right]\bar{L}+\frac{1}{2}{\bar{L}}^{*}\left[X,\bar{L}\right]. \end{array}$$We therefore deduce that the forms in *K* and $K^{\prime }$ will be(21)$$\begin{array}{c} L\left(\cdot \right)=\zeta \left(u,v\right)\;L^{\prime }\left(\cdot \right)=\frac{1}{\gamma (v)}\bar{L}\left(\cdot \right). \end{array}$$This agrees with the expectation that $\bar{L}\;d\tau =L\;dt=L^{\prime }\;dt^{\prime }$. See [7,8,9].

### 2.5. Coherent States

In general, we can consider the incoming quantum noise to be in a coherent state $|\bar{\alpha }\rangle$, where $\bar{\alpha }\left(\tau \right)$ provides the complex amplitude in the rest frame at proper time τ. We obtain this state by displacing the Fock vacuum by(22)$$\begin{array}{c} |\bar{\alpha }\rangle =e^{-∥\bar{\alpha }{∥}^{2}}e^{\int \bar{\alpha }\left(\tau \right){\bar{b}}^{*}\left(\tau \right)\;d\tau }|\mathrm{vac}\rangle . \end{array}$$The normalization involving $∥\bar{\alpha }{∥}^{2}=\int {\left|\bar{\alpha }\left(\tau \right)\right|}^{2}\;d\tau$, which is assumed finite.

In the inertial frame *K*, we will have the complex amplitude(23)$$\begin{array}{c} \alpha \left(t\right)=\frac{1}{\sqrt{\gamma (v)}}\bar{\alpha }\left(\tau \right). \end{array}$$This follows immediately by identifying $\int \bar{\alpha }\left(\tau \right){\bar{b}}^{*}\left(\tau \right)\;d\tau$ with $\int \alpha \left(t\right)b^{*}\left(t\right)\;dt$.

We may replace the vacuum state in (Equation 19) with the coherent state $|\bar{\alpha }\rangle$. The effect of this can be modeled by making the replacements(24)$$\begin{array}{ccc} \bar{L} & \to & {\bar{L}}_{\bar{\alpha }}=\bar{L}+\bar{S}\;\bar{\alpha }\left(\tau \right),\\ \bar{H} & \to & {\bar{H}}_{\bar{\alpha }}=\bar{H}+\frac{1}{2i}\left({\bar{L}}^{*}\bar{\alpha }\left(\tau \right)-\bar{L}{\bar{\alpha }}^{*}\left(\tau \right)\right) \end{array}$$
and averaging in the vacuum state [10]. The instantaneous Lindblad generator is then(25)$$\begin{array}{c} {\bar{L}}_{\bar{\alpha }}\left(X\right)={\bar{\alpha }}^{*}\left(\tau \right)({\bar{S}}^{*}X\bar{S}-X)\bar{\alpha }\left(\tau \right)+{\bar{\alpha }}^{*}\left(\tau \right){\bar{S}}^{*}\left[X,\bar{L}\right]+\left[{\bar{L}}^{*},X\right]\bar{S}\bar{\alpha }\left(\tau \right)+\bar{L}\left(X\right). \end{array}$$We may likewise construct the Lindblad generators $L_{\alpha }$ and $L_{\alpha^{\prime }}^{\prime }$ for the coherent state in *K* and $K^{\prime }$, respectively; however, it is apparent that they scale in exactly the same way as in (Equation 21).

## 3. Uniformly Accelerated Systems

In [3], the situation of rectilinear motion is treated (that is, the system is modeled in a laboratory that has a fixed inertial reference frame). However, the claim is made that the invariance of the quantum stochastic calculus should extend to arbitrary motions as well, with the only complication now being the proper time described by an integral expression.

We argue that this can only be true for situations where the acceleration is negligible. Specifically, the approach must break down if we consider uniformly accelerated systems as here we will additionally encounter the Unruh effect [11,12,13], which predicts that the system will thermalize with inverse (Unruh) temperature(26)$$\begin{array}{c} \beta =\frac{2\pi c}{a\hbar } \end{array}$$
where *a* is the constant proper acceleration.

### Thermal Noise Model

Suppose that our system is a cavity-mode *c* with resonant frequency Ω; then, the thermalization of an oscillator mode is modeled by the quantum Markovian evolution dU(t)=dG(t)U(t) with(27)$$\begin{array}{c} dG\left(t\right)=K⊗dt+\sqrt{\gamma }c⊗dA^{*}\left(t\right)-\sqrt{\gamma }c^{*}⊗dA\left(t\right) \end{array}$$
where γ>0 is a damping constant, $K=-(\frac{1}{2}\left(2n+1\right)\gamma +i\hbar \Omega )c^{*}c$, and the processes $A\left(t\right),A{\left(t\right)}^{*}$ are non-Fock quantum Wiener processes satisfying the quantum Itō table [14](28)$$\begin{array}{c} dA\left(t\right)dA{\left(t\right)}^{*}=\left(n+1\right)\;dt,\;dA{\left(t\right)}^{*}dA\left(t\right)=n\;dt,\\ dA\left(t\right)dA\left(t\right)=0=dA{\left(t\right)}^{*}dA{\left(t\right)}^{*}. \end{array}$$The parameter n>0 and should be set to the average *boson* occupation number for an energy ℏΩ at inverse temperature β:(29)$$\begin{array}{c} n\equiv \frac{1}{e^{\beta \hbar \Omega }-1}. \end{array}$$As is well known, the non-Fock processes may be represented as on a tensor product of a pair of Fock processes. Here, we may employ the Araki–Woods-type construction [15](30)$$\begin{array}{c} A\left(t\right)=\sqrt{n+1}\;B_{1}\left(t\right)⊗I+\sqrt{n}\;I⊗B_{2}{\left(t\right)}^{*}, \end{array}$$
where the $B_{j}$ (j=1,2) are copies of the standard Fock process.

Unfortunately, it is impossible to implement the transformation from the *B* processes to the *A* processes unitarily as the conditions of Shale’s Theorem are not met in this case; see, for instance, [4]. Notably, it is impossible to include an analogue of the counting process Λ(t) in the non-Fock calculus.

## 4. Algebraic Formulation of the Unruh Effect

It is convenient to recast quantum fields in curved spacetimes in the language more familiar to quantum probabilists. For the remainder of this paper, we shall use units where c=1 and ℏ=1. We fix a spacetime (M,g), where M is a four-dimensional pseudo-Riemannian manifold and *g* is metric with signature (+,−,−,−).

We assume the existence of a mapping $\hat{\phi }$ from the set of complex-valued smooth functions of compact support on M into a set of operators. The map $\hat{\phi }$ provides the Klein–Gordon quantization functional and should be *-linear. It should also satisfy $\hat{\phi }\left((□+m^{2})f\right)=0$ and the Einstein causality condition $\left[\hat{\phi }\left(f_{1}\right),\hat{\phi }\left(f_{2}\right)\right]=iE\left(f_{1},f_{2}\right)\;I$, where $E=E^{\mathrm{ret}.}-E^{\mathrm{adv}.}$ is the difference between the retarded and advanced Green’s functions. We shall write $\hat{\phi }\left(f\right)\equiv \int_{M}dt\;dx\;f\left(t,x\right)\;\hat{\varphi }\left(t,x\right)$ and make a standard abuse and refer to $\hat{\phi }\left(\cdot \right)$ over M also as the quantum field.

If $\varphi_{A}$ and $\varphi_{B}$ are complex-valued solutions to the Klein–Gordon equation, then we set ${\left(\varphi_{A},\varphi_{B}\right)}_{\mathrm{KG}}=i\int_{-\infty }^{\infty }dx\;\left\{\varphi_{B}^{*}\partial_{x}\varphi_{A}-\varphi_{A}^{*}\partial_{x}\varphi_{B}\right)\}$. The standard approach is to provide a mode expansion of the quantized field $\hat{\phi }$ in the form(31)$$\begin{array}{c} \hat{\phi }\left(\cdot \right)=\int d^{3}k\;\left(f_{k}\left(\cdot \right)\hat{a}\left(k\right)+f_{k}{\left(\cdot \right)}^{*}\hat{a}{\left(k\right)}^{*}\right) \end{array}$$
where the mode functions $f_{k}$ correspond to positive frequency terms and ${\left(f_{k},f_{k^{\prime }}\right)}_{\mathrm{KG}}=\delta \left(k-k^{\prime }\right)$, ${\left(f_{k}^{*},f_{k^{\prime }}\right)}_{\mathrm{KG}}=0$.

Let A denote the unital *-algebra generated by the $\hat{\phi }\left(f\right)$. More generally, for *R*, a region of M, we denote by A(R) the algebra generated by the $\hat{\phi }\left(f\right)$, with *f* having support in *R*. The Einstein causality condition implies that the algebra $A(R_{1})$ commutes with $A(R_{2})$ whenever the regions $R_{1}$ and $R_{2}$ are causally separated, that is if all vectors from one region to the other are spacelike.

Given a state ω on A, one can construct the corresponding GNS representation $\left(H_{\omega },\pi_{\omega },\Omega_{\omega }\right)$ of A. In the case where M is not compact, then the GNS representations for different ω will typically be unitarily inequivalent. The Minkowski vacuum is the pure state $\omega_{M}$ corresponding to vector $|{\mathrm{vac}}_{M}\rangle$ with the property that it is annihilated by the positive frequency components of the fields; this property is invariant under the Poincaré group and is unique. We will be interested in the Minkowski vacuum state and its restrictions to subalgebras A(R), and these will be gauge-invariant quasi-free (Gaussian) states where the even moments are obtained from sums of pair partitions of the second moments while all odd moments vanish.

Next, let *v* be a complete tangent vector field on M and let $\left\{e^{tv}:t\in R\right\}$ be the one-parameter group of diffeomorphisms it generates. A family of automorphisms $\alpha_{t}$ is then generated on A according to(32)$$\begin{array}{c} \alpha_{t}\left(\hat{\phi }\left(f_{1}\right)\cdots \hat{\phi }\left(f_{n}\right)\right)=\hat{\phi }\left(f_{1}\circ e^{-tv}\right)\cdots \hat{\phi }\left(f_{n}\circ e^{-tv}\right). \end{array}$$We shall be interested in the case where *v* is a time-like Killing vector field, in which case the diffeomorphisms $e^{tv}$ are isometries on the manifold.

We recall the Kubo–Martin–Schwinger (KMS) boundary condition for a state ω, a one-parameter group of isometries $\alpha_{t}$, and a parameter β>0: set $F_{A,B}\left(t_{1},t_{2}\right)=\omega \left(\alpha_{t_{1}}\left(A\right)\alpha_{t_{2}}\left(B\right)\right)$, and then we require $F_{A,B}$ to have an analytic continuation into the region {(z1,z2)∈C2:0≤Im(z2−z1)≤β}, which is bounded and continuous at the boundary of the region, and that(33)$$\begin{array}{c} F_{A,B}\left(t_{1},t_{2}+i\beta \right)=F_{B,A}\left(t_{2},t_{1}\right). \end{array}$$Here, β>0 is the inverse temperature, and ω is said to be a thermal state (technically, the KMS condition here suffices for C*-algebras but not for unbounded operators as is the case here, so a multi-time extension is needed. Fortunately, it suffices for the case of a Gaussian state, and we are restricting our interest to this class [16]).

### 4.1. The Unruh Effect

At this stage, it is worth reviewing the status of the Unruh effect itself. It has been unequivocally established theoretically that an observer moving in a (zero-temperature) vacuum will undergo a convergence to thermal equilibrium at the Unruh temperature. However, the existence of an Unruh radiation was first challenged in [17] and later [18]. The latter paper used standard Markovian approximations from quantum optics (Wigner–Weisskopf); however, this is not an essential objection: it was conclusively shown in the work of Ford and O’Connell [19], in the framework of the quantum Langevin equation, that there is no actual Unruh radiation as such.

We also remark that the Born approximation has been applied to this problem in several recent papers [20,21,22,23,24].

In order to describe a uniformly accelerated observer in Minkowski spacetime with proper acceleration a>0, it is convenient to introduce *radar coordinates*(η,ξ,y,z) given by(34)$$\begin{array}{c} t=\frac{1}{a}e^{a\xi }\sinh\left(a\eta \right),\;x=\frac{1}{a}e^{a\xi }\cosh\left(a\eta \right) \end{array}$$
and one then sees that $dt\;dx=e^{2a\xi }\;d\eta \;d\xi$ and $dT^{2}=e^{2a\xi }\left(d\eta^{2}-d\xi^{2}\right)-dy^{2}-dz^{2}$. The curve ξ,y,z≡0 describes the world line of a particle accelerating with a constant proper acceleration *a*.

Varying the radar coordinates over all real values leads to the region $W_{R}$ given by {(t,x):0<|t|<x} called the *right Rindler wedge* (See Figure 1). The region {(t,x):0<|t|<−x} is the left Rindler wedge, which we denote as $W_{L}$, and this is causally separated from the right wedge. It is obtained from the right wedge by the transformation (t,x)→(−t,−x), and we again define (left) radar coordinates (η,ξ) corresponding to (Equation 34) with minus signs included. In particular, $A(W_{L})$ is the commutant of $A(W_{R})$ and *vice versa.*

The vector field $\frac{\partial }{\partial \eta }=a\left(x\frac{\partial }{\partial t}-t\frac{\partial }{\partial x}\right)$ is a time-like Killing vector field in $W_{R}$. Following Sewell [25], the Unruh effect may now be restated in algebraic terms—specifically as a special case of a Theorem of Bisognano and Wichmann [26,27]. Then, the restriction $\omega_{R}$ of the vacuum state $\omega_{M}$ to A(R) is an invariant under the automorphism group generated by $\frac{\partial }{\partial \eta }$ and moreover satisfies the KMS condition for this group with inverse temperature $\beta =\frac{2\pi }{a}$.

### 4.2. Fields in the Rindler Wedges

We shall adapt the presentation and notations from the book of Birrell and Davies, Section 4.5 in [28]. We restrict to 1 + 1 spacetime and consider the massless Klein–Gordon equation. We may take the mode functions to be(35)$$\begin{array}{c} f_{k}\left(t,x\right)=\frac{1}{\sqrt{4\pi |k|}}\;e^{ikx-i|k|t}. \end{array}$$The k>0 corresponds to right-propagating fields and depends on U=t−x, while k<0 is left-propagating and depends on V=t+x. The coordinates (U,V) are called light-cone coordinates.

The field may be written as $\hat{\phi }\left(\cdot \right)=\int_{-\infty }^{\infty }dk\;\left(f_{k}\left(\cdot \right)\hat{a}\left(k\right)+f_{k}{\left(\cdot \right)}^{*}\hat{a}{\left(k\right)}^{*}\right)$, and we have that $\left[a_{k},a_{k^{\prime }}^{*}\right]=\delta \left(k-k^{\prime }\right)$ with $a_{k}\;|{\mathrm{vac}}_{M}\rangle =0$ for all *k*.

Alternatively, we may introduce modes(36)ukR(η,ξ)=14π|k|eikξ−i|k|ηχR,ukL(η,ξ)=14π|k|eikξ+i|k|ηχL.
where $\chi_{R}$ and $\chi_{L}$ are the characteristic functions for the right and left wedge, respectively. One may then write $\hat{\phi }={\hat{\phi }}_{R}+{\hat{\phi }}_{L}$, where(37)ϕ^R(·)=∫−∞∞dkukR(·)n(|k|)+1d^k(1)+n(|k|)d^k(2)∗+H.c.,ϕ^L(·)=∫−∞∞dkukL(·)n(|k|)+1d^k(2)+n(|k|)d^k(1)∗+H.c.,
with $\left[{\hat{d}}_{k}^{\left(i\right)},{\hat{d}}_{k^{\prime }}^{(j)*}\right]=\delta_{ij}\delta \left(k-k^{\prime }\right)$, and again ${\hat{d}}_{k}^{\left(i\right)}\;|{\mathrm{vac}}_{M}\rangle =0$ for i=1,2 and for all *k*. We also have $n\left(\omega \right)=\frac{1}{e^{2\pi \omega /a}-1}$, which is the mean boson density at the Unruh temperature.

(The exact relation between the $\hat{a}$ fields and the ${\hat{d}}^{\left(i\right)}$ fields is not needed but can be found in [13,28]).

The fields ${\hat{\phi }}_{R}$ and ${\hat{\phi }}_{L}$ have a thermal distribution for the Minkowski vacuum state restricted to the corresponding wedge algebras. We may provide an Araki–Woods representation on the double Fock space H⊗H, where H is the Fock space over the one-particle space $L^{2}\left(R,dk\right)$, with the state being the joint Fock vacuum state Φ⊗Φ. For instance,(38)ϕ^R(·)=B(uR(·))+B(uR(·))∗
where we have the representation(39)$$\begin{array}{c} B\left(f\right)\equiv D^{\left(1\right)}\left(\sqrt{N+1}\;f\right)⊗I^{\left(2\right)}+I^{\left(1\right)}⊗D^{\left(2\right)}{\left(\sqrt{N}\;\bar{f}\right)}^{*} \end{array}$$
with *N* being the operator of multiplication by n(|k|) on $L^{2}\left(R,dk\right)$ and $\bar{f}$ being the complex conjugate of *f* (note $\bar{Nf}=N\bar{f}$). It follows that(40)$$\begin{array}{c} \omega \left(e^{B\left(f\right)+B{\left(f\right)}^{*}}\right)\equiv \langle \Phi ⊗\Phi ,\left(e^{B\left(f\right)+B{\left(f\right)}^{*}}\right)\;\Phi ⊗\Phi \rangle =e^{-\frac{1}{2}\langle f,\left(2N+1\right)f\rangle }. \end{array}$$

## 5. Uniformly Accelerating Detector

We shall adapt the well-known simple model; see for instance the presentation in [28,29] and references therein. We now model a simple detector moving in spacetime. The detector is taken to move along a world line x(τ) parametrized by its proper time τ, although we are interested only in its internal quantum degrees of freedom. The detector just feels the quantum field at its current spacetime location, so we revert to the Dirac picture, where we subtract the free dynamics of both the detector (generated by detector Hamiltonian, $H_{\det.}$, and the field (ultimately described by a family of automorphisms generated by a congruence of integral curves that includes the world line of the detector). The unitary family of relevance to the detector and the local field is taken to be of the form $\frac{d}{d\tau }U_{\lambda }\left(\tau \right)=-i\lambda \Upsilon_{\lambda }\left(\tau \right)\;U\left(\tau \right)$, initialized as $U_{\lambda }\left(0\right)=I_{\det.}⊗I_{\mathrm{field}}$ with(41)$$\begin{array}{c} -i\Upsilon \left(\tau \right)=\left(Le^{-i\Omega \tau }-L^{*}e^{i\Omega \tau }\right)⊗\hat{\phi }\left(x(\tau )\right) \end{array}$$
where *L* is an operator of the detector that is harmonic with frequency Ω under the free dynamics governed by $H_{\det.}$. Here, λ is a coupling parameter, which we shall assume to be limitingly small.

A specific example of this is the Unruh–DeWitt model for a detector [12,30], and here we take it just to be a two-level atom with internal frequency Ω, as measured in its instantaneous rest frame. We take |g〉 and |e〉 to be the ground and excited states of the detector and introduce the lowering operator $\sigma_{-}=\left|g\rangle \langle e\right|$ along with the raising operator $\sigma_{+}=\sigma_{-}^{*}$. In this case, one sets(42)$$\begin{array}{c} H_{\det.}=\frac{1}{2}\Omega \sigma_{z},\;L=\sqrt{\kappa }\;\sigma_{-}. \end{array}$$In other words, we would have the minimal coupling interaction(43)$$\begin{array}{c} -i\Upsilon \left(\tau \right)=\left(\sigma_{-}e^{-i\Omega \tau }-\sigma_{+}e^{i\Omega \tau }\right)⊗\hat{\phi }\left(x(\tau )\right) \end{array}$$Here, the system coupling is through $\sigma_{y}=i\left(\sigma_{-}-\sigma_{+}\right)$, where the lowering and raising terms pick up counter-rotating phases due to $H_{\det.}$. Likewise, we just have the contribution of the field at x(τ).

The detector is assumed to be initially in its ground state with the field in the (Minkowski) vacuum 〈·〉. We note that the two-point function for the field in the Minkowski vacuum state is(44)$$\begin{array}{c} \langle \hat{\phi }\left(x_{1}\right)\hat{\phi }\left(x_{2}\right)\rangle =-\lim_{\epsilon \to 0^{+}}\frac{1}{4\pi^{2}}\frac{1}{{\left(x_{1}-x_{2}-i\epsilon \hat{t}\right)}^{2}} \end{array}$$
where ${\left(x\right)}^{2}\equiv g_{ij}x^{i}x^{j}$, $\hat{t}$ is any future-pointing time-like vector, and the limit is understood in the distributional sense. One may show that for the fields along the world line of the uniform accelerating observer; see, for instance, [28] Equation (3.46),(45)$$\begin{array}{c} \langle \hat{\phi }\left(x(\tau_{1})\right)\hat{\phi }\left(x(\tau_{2})\right)\rangle =-\lim_{\epsilon \to 0^{+}}\frac{a^{2}}{4\pi^{2}}\frac{1}{4\sinh^{2}\left(\frac{a(\tau_{1}-\tau_{2}-i\epsilon )}{2}\right)}. \end{array}$$

The objective is to calculate the probability rate for the detector to transition to its excited state, which should require an absorption of a quantum from the field. Clearly, if the detector moves as an inertial body, this rate should be zero. This changes if the detector is uniformly accelerated, and to this end we take its world line to be $\left(\frac{1}{a}\sinh\left(a\tau \right),\frac{1}{a}\cosh\left(a\tau \right),0,0\right)$ in standard coordinates; this corresponds to η≡τ and ξ=y=z=0 in radar coordinates. The parameter τ will be the proper time along the world line.

### 5.1. Quantum Stochastic Limit for a Uniformly Accelerating Detector

The standard argument at this point is to perform a Fermi Golden Rule calculation; however, we shall frame this as an asymptotic quantum stochastic limit. We introduce a coupling constant λ and rescale proper time as $\tau \to \tau /\lambda^{2}$ (the van Hove rescaling). Then, we take $\lambda \to 0^{+}$. Note that this is a rescaling of the proper time, and here we have the replacement $t\to t=\frac{1}{a}\sinh\left(a\tau /\lambda^{2}\right)\approx \frac{1}{a}e^{a\tau /\lambda^{2}}$ in the fixed inertial frame for τ>0.

With this rescaling, we study $U_{\lambda }\left(\tau /\lambda^{2}\right)$ so that $dU_{\lambda }\left(\tau /\lambda^{2}\right)=-i\Upsilon_{\lambda }\left(\tau \right)\;U_{\lambda }\left(\tau /\lambda^{2}\right)$ with(46)$$\begin{array}{c} -i\Upsilon_{\lambda }\left(\tau \right)=\frac{1}{\lambda }\left(Le^{-i\Omega \tau /\lambda^{2}}-L^{*}e^{i\Omega \tau /\lambda^{2}}\right)⊗\hat{\phi }\left(x(\tau /\lambda^{2})\right). \end{array}$$From this, we obtain the Dyson series expansion(47)$$\begin{array}{c} U_{\lambda }\left(\tau /\lambda^{2}\right)=I_{\det.}⊗I_{\mathrm{field}}+\sum_{n\geq 1}{\left(-i\right)}^{n}\int_{\Delta_{n}\left(\tau \right)}d\tau_{n}\cdots d\tau_{1}\;\Upsilon_{\lambda }\left(\tau_{n}\right)\cdots \Upsilon_{\lambda }\left(\tau_{1}\right) \end{array}$$
where $\Delta_{n}\left(\tau \right)$ is the simplex $\left\{\left(\tau_{n},\cdots ,\tau_{1}\right):t\geq \tau_{n}\geq \cdots \geq \tau_{1}\geq 0\right\}$.

The idea of the quantum stochastic limit [31,32] rests on the observation that, if *g* is an integrable function with $\gamma =\int_{-\infty }^{\infty }g\left(t\right)\;dt$, then(48)$$\begin{array}{c} \lim_{\lambda \to 0^{+}}\frac{1}{\lambda^{2}}\;g\left(\frac{\tau }{\lambda^{2}}\right)=\gamma \;\delta \left(\tau \right). \end{array}$$To see this, we note for any Schwartz test function *f* that $\int_{-\infty }^{\infty }f\left(\tau \right)\frac{1}{\lambda^{2}}g\left(\frac{\tau }{\lambda^{2}}\right)\;d\tau =\int_{-\infty }^{\infty }f\left(\lambda^{2}t\right)g\left(t\right)\;dt\to \gamma \;f\left(0\right)$. Under the correct rescaling, the fields behave like delta-correlated quantum white noises. In our case, we should focus on the rescaled operators $\frac{1}{\lambda }e^{\pm i\Omega \tau /\lambda^{2}}\hat{\phi }\left(x(\tau /\lambda^{2})\right)$.

We consider the rescaled two-point function(49)$$\begin{array}{c} \frac{1}{\lambda^{2}}\;e^{-i\Omega \left(\tau_{1}-\tau_{2}\right)/\lambda^{2}}\langle \hat{\phi }\left(x(\tau_{1}/\lambda^{2})\right)\hat{\phi }\left(x(\tau_{2}/\lambda^{2})\right)\rangle \to \gamma \;\delta \left(\tau_{1}-\tau_{2}\right) \end{array}$$
with(50)$$\begin{array}{c} \gamma =-\lim_{\epsilon \to 0^{+}}\int_{-\infty }^{\infty }\frac{a^{2}}{4\pi^{2}}\frac{e^{-i\Omega \tau }\;d\tau }{4\sinh^{2}\left(\frac{a(\tau_{1}-\tau_{2}-i\epsilon )}{2}\right)}\equiv \frac{\Omega }{4\pi }\;n\left(\Omega \right) \end{array}$$
where $n\left(\Omega \right)=\frac{1}{e^{2\pi \Omega /a}-1}$, and the integral may be evaluated by the residue theorem.

Using the form (Equation 37) and setting $\eta =\tau /\lambda^{2},\;\xi =0$ to yield the locus of the world line, we may write(51)$$\begin{array}{ccc} \frac{1}{\lambda }\;e^{+i\Omega \tau /\lambda^{2}}{\hat{\phi }}_{R}\left(x(\tau /\lambda^{2})\right) & = & \frac{1}{\lambda }\int_{-\infty }^{\infty }dk\;e^{i\left(\Omega -|k|\right)\tau /\lambda^{2}}\left(\sqrt{n(|k|)+1}\;{\hat{d}}_{k}^{\left(1\right)}+\sqrt{n(|k|)}\;{\hat{d}}_{k}^{(2)*}\right)\\ & + & \frac{1}{\lambda }\int_{-\infty }^{\infty }dk\;e^{i\left(\Omega +|k|\right)\tau /\lambda^{2}}\left(\sqrt{n(|k|)+1}\;{\hat{d}}_{k}^{(1)*}+\sqrt{n(|k|)}\;{\hat{d}}_{k}^{\left(2\right)}\right). \end{array}$$In the limit λ→0, the first of these terms will pick up a resonance for the mass-shell |k|=Ω>0; the second will lead to rapid oscillations and may be ignored. We therefore may expect the following asymptotic limits (see Section 6 for precise details)(52)$$\begin{array}{c} \frac{1}{\lambda }e^{-i\Omega \tau /\lambda^{2}}\;\hat{\phi }\left(x(\tau /\lambda^{2})\right)\to \sqrt{\frac{\Omega }{4\pi }}\;a{\left(\tau \right)}^{*},\\ \frac{1}{\lambda }\;e^{+i\Omega \tau /\lambda^{2}}\hat{\phi }\left(x(\tau /\lambda^{2})\right)\to \sqrt{\frac{\Omega }{4\pi }}\;a\left(\tau \right). \end{array}$$
where the limit processes are (Gaussian) thermal quantum white noises satisfying(53)$$\begin{array}{c} \left[a\left(\tau_{1}\right),a{\left(\tau_{2}\right)}^{*}\right]=\delta \left(\tau_{1}-\tau_{2}\right), \end{array}$$
with(54)$$\begin{array}{c} \langle a\left(\tau_{1}\right)a{\left(\tau_{2}\right)}^{*}\rangle =\left(n(\omega )+1\right)\;\delta \left(\tau_{1}-\tau_{2}\right),\;\langle a{\left(\tau_{1}\right)}^{*}a\left(\tau_{2}\right)\rangle =n\left(\omega \right)\;\delta \left(\tau_{1}-\tau_{2}\right), \end{array}$$
along with $\langle a\left(\tau_{1}\right)a\left(\tau_{2}\right)\rangle =0=\langle a{\left(\tau_{1}\right)}^{*}a{\left(\tau_{2}\right)}^{*}\rangle$.

We should look at the integrated versions of these quantum white noises, formally $A\left(\tau \right)=\int_{0}^{\tau }b\left(\sigma \right)\;d\sigma$, and these are non-Fock quantum Wiener processes with quantum Itō table(55)$$\begin{array}{c} dAdA^{*}=\left(n(\Omega )+1\right)\;dt,\;dA^{*}dA=n\left(\Omega \right),\;dt\;dAdA=dA^{*}dA^{*}=0. \end{array}$$

The limit evolution operator for $U_{\lambda }\left(\tau /\lambda^{2}\right)$ will be given by the quantum stochastic process satisfying the formal quantum white noise differential equation(56)$$\begin{array}{c} \frac{d}{dt}U\left(t\right)=\sqrt{\kappa \frac{\Omega }{4\pi }}\left(L⊗a{\left(\tau \right)}^{*}-L^{*}⊗a\left(\tau \right)\right)U\left(t\right) \end{array}$$
and this corresponds to the non-Fock quantum stochastic differential equation [14](57)$$\begin{array}{ccc} dU(\tau ) & = & (\sqrt{\Gamma }\;\sigma_{-}⊗dB{\left(\tau \right)}^{*}-\sqrt{\Gamma }\;\sigma_{+}⊗dB\left(\tau \right)\\ & & -\frac{1}{2}\Gamma \left(n(\Omega )+1\right)\sigma_{+}\sigma_{-}⊗d\tau -\frac{1}{2}\Gamma n\left(\Omega \right)\sigma_{-}\sigma_{+}⊗d\tau -iH_{\det.}^{\prime }⊗d\tau )U\left(t\right). \end{array}$$We have introduced the coupling strength $\Gamma =\frac{\kappa \Omega }{4\pi }$. We also obtain a *Lamb-shift*-type correction to the detector Hamiltonian: $H_{\det.}^{\prime }=\kappa \left(\varepsilon_{-}L^{*}L+\varepsilon_{+}LL^{*}\right)$, where(58)$$\begin{array}{c} \frac{1}{2}\gamma +i\varepsilon_{\pm }=-\lim_{\epsilon \to 0^{+}}\int_{-\infty }^{0}\frac{a^{2}}{4\pi^{2}}\frac{e^{\pm i\Omega \tau }\;d\tau }{4\sinh^{2}\left(\frac{a(\tau_{1}-\tau_{2}-i\epsilon )}{2}\right)}. \end{array}$$

Note that we may represent the non-Fock processes on the doubled-up Fock space with $A\left(\tau \right)\equiv \sqrt{n(\Omega )+1}B_{1}\left(\tau \right)⊗I_{2}+\sqrt{n(\Omega )}I_{1}⊗B_{2}{\left(\tau \right)}^{*}$, with the joint vacuum state as cyclic state.

This describes the detector as an open system with Lindblad generator(59)$$\begin{array}{c} L\left(X\right)=\left(n(\Omega )+1\right)\;\Gamma L_{\sigma_{+}\sigma_{-}}\left(X\right)+n\left(\Omega \right)\;\Gamma L_{\sigma_{-}\sigma_{+}}\left(X\right)-i\left[X,H_{\det.}\right], \end{array}$$
where $L_{L}\left(X\right)=\frac{1}{2}\left[L^{*},X\right]L+\frac{1}{2}L^{*}\left[X,L\right]$.

The state of the detector may be described by a density matrix ρ(τ), where $\mathrm{tr}\left(\rho \left(\tau \right)\;X\right)=\langle \mathrm{tr}(\rho \;U{\left(\tau \right)}^{*}\left(X⊗I\right)U\left(\tau \right)\rangle$ for any detector-observable *X*. From this, we obtain the master equation $\frac{d}{d\tau }\rho \left(\tau \right)=L^{★}\rho \left(\tau \right)$, where $L^{★}$ is the Liouvillian defined as the adjoint of the Lindbladian through the duality $\mathrm{tr}\left(L^{★}\rho \;X\right)=\mathrm{tr}\left(\rho \;L\left(X\right)\right)$.

**Remark** **2.**
*A gauge process Λ(τ) cannot be defined for non-Fock quantum stochastic calculus.*


### 5.2. Unruh–DeWitt Detector

In the special case of the Unruh–DeWitt detector where the model is a two-level atom and $L\propto \sigma_{-}$, it is easy to see that ρ(τ) will relax to the thermal state $\rho_{\mathrm{thermal}}=n_{F}\left(\Omega \right)\sigma_{-}\sigma_{+}+\left(1-n_{F}\left(\Omega \right)\right)\sigma_{+}\sigma_{-}$. Here, $n_{F}\left(\Omega \right)=\frac{n(\Omega )+1}{2n(\Omega )+1}=\frac{1}{e^{2\pi \Omega /a}+1}$ is the *fermion* average occupation number at thermal equilibrium (as expected for a two-level atom consisting of ground and excited states). In fact, we solve the master equation exactly to obtain(60)$$\begin{array}{ccc} \langle e|\rho (\tau )|e\rangle & = & \frac{n(\Omega )}{2n(\Omega )+1}+\left(\langle e|\rho \left(\tau \right)|e\rangle -\frac{n(\Omega )}{2n(\Omega )+1}\right)e^{-(2n(\Omega )+1)\Gamma \tau },\\ \langle g|\rho (\tau )|g\rangle & = & \frac{n(\Omega )+1}{2n(\Omega )+1}+\left(\langle g|\rho \left(\tau \right)|g\rangle -\frac{n(\Omega )+1}{2n(\Omega )+1}\right)e^{-(2n(\Omega )+1)\Gamma \tau },\\ \langle e|\rho (\tau )|g\rangle & = & \langle e|\rho \left(\tau \right)|g\rangle e^{-(2n(\Omega )+1)\Gamma \tau /2}. \end{array}$$

**Remark** **3.**
*We could alternatively consider a cavity-mode c detector, where $H_{\det.}=\Omega c^{*}c$ with $L=\sqrt{\gamma }\;c$, in which case we obtain the (Equation 27).*


## 6. Formulation of the Limit

We introduce the processes $A_{\lambda }{\left(\tau \right)}^{*}=\frac{1}{\sqrt{\gamma }}\int_{0}^{\tau }\frac{1}{\lambda }e^{-i\Omega \sigma /\lambda^{2}}\hat{\phi }\left(x(\sigma /\lambda^{2})\right)$ and $A_{\lambda }\left(\tau \right)=\frac{1}{\sqrt{\gamma }}\int_{0}^{\tau }\frac{1}{\lambda }e^{+i\Omega \sigma /\lambda^{2}}\hat{\phi }\left(x(\sigma /\lambda^{2})\right)$. From (Equation 49), we see that(61)$$\begin{array}{c} \lim_{\lambda \to 0}\left[A_{\lambda }\left(\tau \right),A_{\lambda }{\left(\sigma \right)}^{*}\right]=\min\left(\tau ,\sigma \right). \end{array}$$

For *f*, a square-integrable function, we consider $A{\left(f,\lambda \right)}^{*}=\int_{0}^{\infty }f\left(\tau \right)dA_{\lambda }{\left(\tau \right)}^{*}$ and $A\left(f,\lambda \right)=\int_{0}^{\infty }f{\left(\tau \right)}^{*}dA_{\lambda }\left(\tau \right)$. We consider the Weyl unitaries $W\left(f,\lambda \right)=\exp\{A{\left(f,\lambda \right)}^{*}-A\left(f,\lambda \right)\}$. We then have(62)$$\begin{array}{c} \lim_{\lambda \to 0}\omega \left(W(f,\lambda )\right)=e^{-\frac{1}{2}\left(2n\left(\Omega \right)+1\right)\int_{0}^{\infty }d\tau \;{\left|f\left(\tau \right)\right|}^{2}}. \end{array}$$

We interpret the limits of these processes as non-Fock quantum stochastic processes with a gauge-invariant quasi-free (Gaussian) state. This may be naturally represented using the Araki–Woods construction as processes(63)$$\begin{array}{c} A\left(\tau \right)=\sqrt{n(\Omega )+1}\;D_{1}\left(\tau \right)⊗I_{2}+\sqrt{n(\Omega )}I_{1}⊗D_{2}{\left(\tau \right)}^{*}, \end{array}$$
where $D_{1}$ and $D_{2}$ are copies of the standard Fock quantum stochastic processes, with the state being the joint Fock vacuum Ψ=Φ⊗Φ.

For f,g square-integrable, we define(64)$$\begin{array}{c} \omega_{f,g}\left(Z\right)=\omega \left(W{\left(f,\lambda \right)}^{*}\;Z\;W\left(g,\lambda \right)\right). \end{array}$$

For $\psi ,\psi^{\prime }$ vectors on the detector space, we understand $\langle \psi ,\omega_{f,g}^{\left(\lambda \right)}\left(\cdot \right)\;\psi^{\prime }\rangle$ to be the linear functional on the joint algebra of the detector and field, given by(65)$$\begin{array}{c} \langle \psi ,\omega_{f,g}^{\left(\lambda \right)}\left(X⊗Z\right)\;\psi^{\prime }\rangle =\langle \psi ,X\;\psi^{\prime }\rangle \;\omega_{f,g}\left(Z\right). \end{array}$$

We shall say that a family of operators $Q_{\lambda }$ on the detector + right Rindler field converges **weakly in collective coherent states** to *Q* if(66)$$\begin{array}{c} \lim_{\lambda \to 0}\langle \psi ,\omega_{f,g}^{\left(\lambda \right)}\left(Q_{\lambda }\right)\;\psi^{\prime }\rangle =\langle \psi ⊗W_{A}\left(f\right)\Psi ,Q\;\psi^{\prime }⊗W_{A}\left(g\right)\Psi \rangle , \end{array}$$
for all detector states $\psi ,\psi^{\prime }$ and all square-integrable test functions f,g. Here, $W_{A}\left(f\right)$ denotes the corresponding Weyl unitary $W_{A}\left(f\right)=\exp\left\{A{\left(f\right)}^{*}-A\left(f\right)\right\}$.

The precise statement is that $U_{\lambda }\left(\tau /\lambda^{2}\right)$ converges weakly to the quantum stochastic process U(τ) on the limiting non-Fock noise space. In fact, one also has the convergence in the same mode of $U_{\lambda }{\left(\tau /\lambda^{2}\right)}^{*}\left(X⊗I\right)U_{\lambda }\left(\tau /\lambda^{2}\right)$ to $U{\left(\tau \right)}^{*}\left(X⊗I\right)U\left(\tau \right)$ for each bounded operator *X* of the detector.

The proof of convergence follows essentially the same lines as [33]. The first step is to use the Araki–Woods representation for the right Rindler field given by Equations (Equation 38) and (Equation 39). We first take the Dyson series expansion (Equation 47) and examine the functional $\langle \psi ,\omega_{f,g}^{\left(\lambda \right)}\left(\cdot \right)\;\psi^{\prime }\rangle$ on the *n* term in the series. Allowing the Weyl unitaries to displace the fields and collecting terms together as polynomials in the fields, we see that the field expectations of an even number of terms can be written in terms of the two-point functions (the odd terms vanishing!) due to the mean zero quasi-free nature of the Minkowski vacuum state. These come in two types: type I, where the pair contractions are time-consecutive with respect to the simplicial ordering; and type II, where there is at least one non-consecutive contraction. One can readily show that each type II term makes zero contribution as λ→0. Moreover, one can show that the series is absolutely summable, meaning that only type I terms contribute. This is in effect the Markovian feature.

**Remark** **4.**
*Note that, in the usual treatment of the Unruh–DeWitt detector, it is argued that, since $L\propto \sigma_{-}$, we have $L^{2}=L^{*2}=0$, which fortuitously kills off the type II terms. However, for the quantum stochastic limit to hold, we do not need this property, and our only assumption is that L is harmonic under the free dynamics. The type II terms disappear in the limit as λ→0, and we obtain a Markovian description. As such, the limit has much wider validity than just the special case of a two-level atom with raising/lowering coupling.*


The time-consecutive contractions are delta-correlated in the limit, and we obtain a 1/2-range contribution due to the simplicial ordering. This is why we encounter the factors (Equation 58). The surviving type I terms can be re-summed to yield the matrix elements for the corresponding limit U(t), solving (Equation 57).

## 7. Discussion and Conclusion

We have re-derived the transformations of the quantum stochastic evolutions (Equation 15) presented in [3] but in a more direct manner. In their treatment, they consider 1 + 1 dimensional spacetime and have two bosonic fields: one traveling from the right and one from the left. The multiple-field case does not pose any problem as the same transformations, (Equation 12) and (Equation 15), apply to all the scattering, annihilation, creation, and time processes.

In principle, we could even handle the usual 1 + 3 spacetime by introducing a bosonic noise for each unit vector in 3-space. This would require us to take the multiplicity space for the noise to be infinite-dimensional, but this is inherent in the Hudson and Parthasarathy formalism [1].

The transformation rules for the quantum stochastic calculus apply to systems that move as inertial systems as described by inertial observers. If the system is slowly accelerating, then we may expect this to still be approximately true; however, this cannot be the case in general, as illustrated by the Unruh effect for uniformly accelerated systems. The Unruh effect essentially only says that a uniformly accelerated observer finds that the quantized Klein–Gordon field in the Minkowski vacuum state acts to thermalize the observer at the Unruh temperature (proportional to the proper acceleration); it does not, strictly speaking, make any statement about the existence of radiation detectable as particles. The non-existence of the gauge process Λ(τ) is consistent with the claim that there are no Unruh particles as such.

The quantum stochastic calculus was developed by Hudson and Parthasarathy as a method of dilating quantum dynamical semi-groups, and their view would have been that the dilation is a purely mathematical construct. In the present context, this works well with the algebraic view: in particular, there is no onus on interpreting a gauge process as counting the number of particles. The later version due to Gardiner and Collett emphasizes the input–output formalism and is more rooted in physical terms.

It should be noted that we have obtained an open system description of a quantum mechanical system undergoing a uniform acceleration as a van Hove limit. This gives us a Markovian model that shows relaxation to the thermal state at the corresponding Unruh temperature. Detector models, such as the Unruh–DeWitt detector, are traditionally analyzed in terms of transition rates, being naturally interpreted as the detector conducting measurements on the field. However, for quantum Markovian systems, it is more useful to develop an input–output approach where the scattered output field is measured (typically a homodyne measurement) and derive a stochastic master equation for the detector. In this situation, the detector is not actually measured directly; instead, we infer its state based on measurements of, say, an output quadrature. The question of quantum filtering based on measuring a quadrature of the output field is an open one; however, we have taken a key step in this direction by obtaining a quantum Markov model for the system and noise.

## Figures and Tables

**Figure 1 entropy-27-00529-f001:**
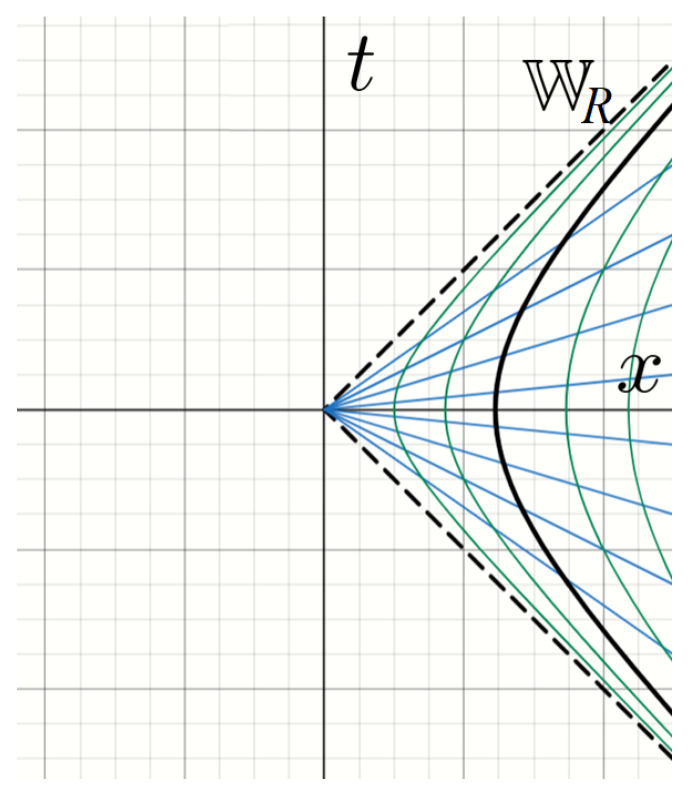
The Rindler wedge $W_{R}$. Curves of constant τ (blue) and constant ξ (green) are plotted. The special case ξ=0 provides the world line of a particle accelerating at fixed proper acceleration a>0 (bold).

## Data Availability

No new data were created or analyzed in this study.

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
