# Peer review of "A Note on the Relativistic Transformation Properties of Quantum Stochastic Calculus"

_entropy, 2025, doi:10.3390/e27050529_

Round 1

Reviewer 1 Report

Comments and Suggestions for Authors

  Quantum stochastic calculus is an important research field in the intersection of quantum mechanics and stochastic processes. It holds certain theoretical value it by combining it with the transformation properties of relativity and exploring the situation in acceleration systems. Particularly, in this research, the authors provides new perspectives and methods for understanding the behavior of quantum systems in relativistic environments and aspects such as the Unruh effect. The paper involves cutting-edge concepts such as quantum open system dynamics and quantum noise. These studies have a certain promoting effect on the development of fields like quantum information science and quantum field theory. However, there are some unclear aspects and drawbacks in the current manuscript. For an overall improvement of the manuscript, I would suggest the authors further address the following points: 

  1. Some of the references are relatively outdated and do not cover the latest research achievements in this field. The references cited in themanuscript mainly focus on research achievements from the 1970s to the 1980s. It is recommended to include more recent research developments to enhance the timeliness and relevance of the paper to current research areas. Moreover, when citing references, it is not clearly indicated which viewpoints or methods from specific references were adopted, making it difficult for readers to judge the innovation of the research and its relationship with previous works.
  2. The derivationsof some formulas are rather discontinuous. For instance, in the derivation of ζ(u,v) from the Lorentz transformation and the transformation rules of quantum stochastic processes, the mathematical basis for some key steps is not expounded in detail. Moreover, in the discussion of the limit process, the proofs of some mathematical properties are not sufficiently profound, such as the convergence of the limit.
  3. For some important physical concepts, such as quantum white noise and non-Fock quantum random processes, there is a lack of intuitive physical image explanations. When discussing the Unruh effect, the elaboration on its essence and physical significance is not profound enough, and the manifestation of this effect in actual physical phenomena has not been fully explained.
  4. In Section 4, there is a spelling error: "should ne set" should be corrected to "should be set." To improve the accuracy and readability of the manuscript, it is recommended that the authorsconduct a thorough proofreading of the entire text.
  5. It is recommended to add textual explanations after key equations to clarify the meaning of their symbols, thereby enhancing the clarity of the formulas. For example, in Eq.(31), the authors did not explicitly explain the physical meanings of the symbols B1(t) and B2(t).
  6. The manuscriptfocuses on fundamental theory but does not address potential application areas. It is recommended to include an outlook on the application prospects in the conclusion section.

  In summary, there are many unclear aspects and drawbacks that demand further improvement and clarification in the current paper. I suggest the authors make comprehensive revisions to enhance the quality of the manuscript, such as enhancing the rigor of mathematical derivations, clearly explaining physical concepts, optimizing the structure of the paper, updating the references and clarifying the citation relationships, carefully considering all the comments raised above. I can recommend the paper for publication if it meets the acceptance criteria of Entropy.

Author Response

Please find attached our replies to your valuable comments.

Reviewer 2 Report

Comments and Suggestions for Authors

In this manuscript under review, the author presents a simple argument to derive the relativistic transformation of the quantum stochastic calculus formalism between inertial observers, 
and to obtain the quantum open system dynamics for a system moving in a coherent (including vacuum) quantum field under the usual Markov approximation.
The arguments proposed in this manuscript provide a unifying approach to Bose and Fermi systems, as well as to the Araki-Woods type construction, which can also be applied to explain the Unruh effect.
This manuscript is well written and quite interesting, and therefore recommended for publication in Entropy.

Author Response

(The authors gave the same response as above.)

Round 2

Reviewer 1 Report

Comments and Suggestions for Authors

The author has adequately addressed all my comments. The revised manuscript is much clearer than its previous version. Therefore, I recommend the publication of the manuscript in its present form.